# Domain Organization in Plant Blue-Light Receptor Phototropin2 of *Arabidopsis thaliana* Studied by Small-Angle X-ray Scattering

**DOI:** 10.3390/ijms21186638

**Published:** 2020-09-10

**Authors:** Masayoshi Nakasako, Mao Oide, Yuki Takayama, Tomotaka Oroguchi, Koji Okajima

**Affiliations:** 1Department of Physics, Faculty of Science and Technology, Keio University, 3-14-1 Hiyoshi, Kohoko-ku, Yokohama, Kanagawa 223-8522, Japan; mao@z3.keio.jp (M.O.); takayama@sci.u-hyogo.ac.jp (Y.T.); oroguchi@phys.keio.ac.jp (T.O.); okajima.k.502@gmail.com (K.O.); 2RIKEN SPring-8 Center, 1-1-1 Kouto, Sayo-cho, Sayo-gun, Hyogo 679-5148, Japan; 3Graduate School of Material Science, University of Hyogo, 3-2-1 Kouto, Kamigori-cho, Ako-gun, Hyogo 678-1297, Japan

**Keywords:** phototropin2, small-angle X-ray scattering, photoreceptor protein, signal transduction

## Abstract

Phototropin2 (phot2) is a blue-light (BL) receptor protein that regulates the BL-dependent activities of plants for efficient photosynthesis. Phot2 is composed of two light-oxygen-voltage sensing domains (LOV1 and LOV2) to absorb BL, and a kinase domain. Photo-activated LOV domains, especially LOV2, play a major role in photo-dependent increase in the phosphorylation activity of the kinase domain. The atomic details of the overall structure of phot2 and the intramolecular mechanism to convert BL energy to a phosphorylation signal remain unknown. We performed structural studies on the LOV fragments LOV1, LOV2, LOV2-linker, and LOV2-kinase, and full-length phot2, using small-angle X-ray scattering (SAXS). The aim of the study was to understand structural changes under BL irradiation and discuss the molecular mechanism that enhance the phosphorylation activity under BL. SAXS is a suitable technique for visualizing molecular structures of proteins in solution at low resolution and is advantageous for monitoring their structural changes in the presence of external physical and/or chemical stimuli. Structural parameters and molecular models of the recombinant specimens were obtained from SAXS profiles in the dark, under BL irradiation, and after dark reversion. LOV1, LOV2, and LOV2-linker fragments displayed minimal structural changes. However, BL-induced rearrangements of functional domains were noted for LOV2-kinase and full-length phot2. Based on the molecular model together with the absorption measurements and biochemical assays, we discuss the intramolecular interactions and domain motions necessary for BL-enhanced phosphorylation activity of phot2.

## 1. Introduction

Phototropin (phot), which is found in green algae to land plants, is a blue-light (BL) receptor protein that regulates the adaptation of organisms for efficient photosynthesis [1,2]. *Chlamydomonas* phot mediates feedback regulation of photosynthesis via gene expression [3]. Photo-mediated responses in land plants include phototropism [4,5], chloroplast movement [5,6,7], stomata opening [8], and leaf expansion [9]. Most higher plants have two isoforms of phot, designated phot1 and phot2. The isoforms have different sensitivities to BL intensity. Phot1 operates those responses over a broad range of BL intensities, while phot2 is activated under high fluence irradiation [5,10]. Therefore, while BL-activated phot1 and phot2 redundantly regulate phototropism, accumulation response of chloroplasts, and stomata opening under moderate irradiation, phot2 alone mediates the avoidance response of chloroplasts under high fluence irradiation [6]. Despite the homologous primary structures, the structural basis for the different BL-sensitivities is still unclear.

Phot2 from *Arabidopsis thaliana* comprises 915 amino acid residues and two flavin mononucleotide (FMN) molecules [11] (Figure 1a). The polypeptide chain folds into three functional domains displaying small disorder properties (Figure 1b). Two light-oxygen-voltage domains (designated LOV1 and LOV2 from the *N*-terminus) [12,13] reside in the *N*-terminal half. A serine/threonine kinase domain (STK) is in the *C*-terminal half. LOV is a member of the Per-Arnt-Sim superfamily [14]. In each LOV domain, one FMN molecule is non-covalently bound to a pocket formed in a characteristic arrangement of α-helices and β-strands (α/β-scaffold; Figure 1c,d) [12,15]. The LOV domains of phots from various organisms have similar α/β-scaffold structures and similar binding modes of FMN molecules, as revealed by crystal structure analyses of LOV2 of *Adiantum* phy3 [15], LOV1 of *Chlamydomonas* phot [16], LOV2 of *Avena* phot1 [17], LOV1 of *Arabidopsis* phot1 and phot2 [18], LOV2 of *Arabidopsis* phot2 [19], and LOV2 of *Arabidopsis* phot1 [20].

Upon BL irradiation, the LOV domain in the dark state (designated D_450_) undergoes the characteristic photo-cycle (Figure 1e) [21,22]. The absorption spectrum is significantly changed in the S_390_ state (Figure 1f) [21,22], in which an adduct of FMN is transiently formed with a sulfur atom of a well-conserved cysteine residue (for instance, Cys170 in LOV1 and Cys426 in LOV2 of phot2, as shown in Figure 1a) [11,12,23]. Small and local structural changes in the adduct form induce reorganization of hydrogen bonds between the FMN and sidechains of amino acid residues forming the pocket [16,23]. S_390_ thermally reverts to D_450_ with a time constant ranging from seconds to minutes in the dark [24].

STK is classified into group VIII of the AGC protein kinase family [25]. Since BL irradiation enhances the phosphorylation activity of STK, phot2 converts the photon energy absorbed by LOV to conformational changes by intramolecular mechanisms to control the activity of STK [1]. Since the amino acid sequence of STK is homologous to that of a cAMP-dependent protein kinase [26], the three-dimensional structure of STK can be predicted from the crystal structure of the kinase (Figure 1c) [22].

Regarding the interdomain interactions necessary to convert BL photon energy to the phosphorylation, a rescue experiment for phot2 deficient *A. thaliana* demonstrated that LOV2 alone can enhance the phosphorylation activity of STK under BL irradiation [27,28]. Therefore, the LOV2-STK fragment is believed to be the minimum functional unit necessary to induce macroscopic BL responses in *Arabidopsis*. Structural studies on LOV2-STK fragments of *Arabidopsis* phot1, phot2, and *Chlamydomonas* phot reported a common regulation mechanism to control the phosphorylation activity of STK [22,29,30]. On the other hand, a study on *Chlamydomonas* phot suggested the functional role of LOV1 in the photosensitivity of STK activation [30].

The regions connecting the functional domains display high scores of natural unfolding, as the *N*- and *C*-terminal regions [22] (Figure 1b). In particular, the *N*-terminal region between LOV2 and STK in *Avena* phot1 folds into an amphipathic α-helix, designated Jα, which associates with the C-terminal lateral region of the α/β-scaffold of LOV2 [17,31]. The Jα-helix displays BL-induced conformational changes, folding-unfolding transition, and dissociation from the LOV2 domain [31,32,33]. LOV2-Jα of *Arabidopsis* phot2 is assumed to have a helical structure that is similar to that of the Jα-helix of *Avena* phot1 LOV2 (Figure 1c). Unfolding and conformational changes of the Jα-helix are thought to be essential steps for intramolecular signaling from the LOV2 to STK domain [34].

As a result of the functional ensemble among LOVs, STK, and the linker region, BL-activated phot2 displays auto-phosphorylation, and also phosphorylates proteins in the downstream part of the signaling pathway [35,36,37]. Therefore, phot2 can convert physical BL stimuli to biochemical phosphorylation signals that activate signal transduction cascades that are essential for BL-induced responses in plants. To understand the molecular mechanism of this conversion, structural studies on phot in D_450_ and S_390_ are essential.

We commenced structural studies in 2002 with the goal of understanding the molecular mechanism of phot2 in A. thaliana. These early studies mainly focused on the structures of recombinant fragments containing LOV domains due to the difficulties in expression and purification of full-length phot2 [22,38]. In addition, since crystallization of purified fragments failed for recombinant specimens, except LOV1 [18,39], structures of fragments and full-length phot2 were investigated using small-angle X-ray scattering (SAXS) (Figure 2a) [22,38,40].

SAXS of a protein in solution provides a one-dimensional intensity profile (Figure 2b), which is averaged over the conformational ensemble and orientation against the direction of the incident X-ray. Structural information obtained from the profile includes the radius of gyration (*R*g) (Figure 2c), the maximum dimension of the molecule, and the molecular shapes at low resolution. Although SAXS provides little structural information of proteins at an atomic resolution, it is valuable in visualizing low-resolution structures of proteins in solution [46,47], which facilitates investigations of the dynamics of proteins in solution [48], and monitoring the structural changes induced by light stimuli (Figure 2a) [49]. In addition, SAXS combined with size-exclusion chromatography (SEC) has extended its capability even for specimens displaying aggregation at high concentrations [50,51,52].

Three-dimensional structures of proteins are very difficult to retrieve from SAXS profiles because these profiles are averaged over both the variation in protein conformations and the orientation of proteins against the direction of the incident X-rays during exposure. To overcome these difficulties, ab initio algorithms have been developed [45,53,54,55], and have also contributed to the vast applications in structural studies of proteins in solution [56]. However, ab initio calculations do not provide a unique structure model, and nonrealistic models can result in calculation trials due to the lack of information in the SAXS profiles. Various attempts to evaluate restored models have been made [57,58]. However, nonrealistic models blurred the details of a model averaged from all restored models. To overcome this problem, we developed a protocol to objectively extract probable models by multivariate analysis and classification for restored models [59] (Figure 2d). This protocol can suggest sets of probable models suitable for discussing structures of proteins together with biochemical and other structural information [43,52].

Here, we review our SAXS studies on recombinant LOV1, LOV2 domains, LOV2-linker [41], LOV2-STK fragments [22], and full-length [43] of phot2 from A. thaliana, including the developed SAXS analysis method [56]. Based on the SAXS parameters and models, we discuss the arrangement of domains in full-length phot2, and the intramolecular mechanism to transmit and amplify small structural changes in BL-activated LOV domains to STK to enhance phosphorylation activity.

## 2. Results

### 2.1. Photochemical Properties

Recombinant LOV-containing fragments and full-length phot2 in the dark displayed absorption spectra of D_450_ comprising three absorption peaks at 474, 446, and 371 nm, and a shoulder at approximately 410 nm (Figure 1f). The ratio between the absorbance values at 273 nm and 446 nm in D_450_ was used to estimate the number of FMN molecules bound to each expressed fragment or full-length phot2 is estimated [22,41,43].

Under BL irradiation, the LOV domains of the recombinant specimens were converted to the S_390_ state. The accumulated S_390_ in each specimen estimated from the absorption spectra, depended on the fluence rate. Prior to the SAXS experiments, the fluence rate and irradiation time necessary for the conversion of almost all molecules in the specimen solution to S_390_ were determined. In addition, the time necessary for dark reversion was also measured spectroscopically. In the LOV2-STK fragment and full-length phot2, the Asp720Asn mutation introduced to avoid auto-phosphorylation of STK (see Materials and Methods section) had little influence on the absorption spectra of both D_450_ and S_390_, and on the kinetics of the D_450_-S_390_ conversion and S_390_-D_450_ dark reversion [22,43].

### 2.2. LOV1, LOV2, and LOV2-Linker Fragments

Figure 3a depicts the SAXS profiles of LOV1, LOV2, and LOV2-linker (Figure 1a) in the dark and under BL irradiation. First, the SAXS profiles of each fragment were almost the same between the D_450_ and S_390_ states, indicating that conformational changes between the two states were too small to detect in SAXS. The fragments were monodispersive as indicated by the Guinier plots (see the Materials and Methods section), which was approximated by single regression lines (Figure 3b and Table 1) and the linear concentration dependency of both *C*/*I*(*S*,*C*) [41] and *R*g(C)^2^ (see the Materials and Methods section and Figure 3c).

The SAXS profile of LOV1 (Figure 3a) suggested a larger molecular size than LOV2 because of the difference in their association modes, as described below. The inconsistent profile of the LOV2-linker with that of LOV2 indicated that, rather than being a flexible loop, the compact folding of the linker significantly contributed to the SAXS profile. The estimated apparent of LOV2 and LOV2-linker indicated that each was a monomer in solution, while LOV1 was a dimer (Table 1). The signs of parameters *A*_2_ and *B*_if_ in the Materials and Methods section for each fragment were unchanged, and their D_450_ and S_390_ values were similar to one another. These results were consistent with the idea that BL-induced changes in the surface properties of either fragment would be undetectably small in the concentration dependencies of *C*/*I*(*S*,*C*) and *R*g(C)^2^ (Figure 3c).

All restored molecular models of LOV2 were similar to each other and displayed an anisotropic shape (Figure 3d). The crystal structure of LOV2 from *Arabidopsis* phot2 [19] fit the main body of the averaged SAXS model (Figure 3d). The LOV2-linker model displayed an elongated extension from a main body assignable to LOV2 (Figure 3d). Therefore, the linker region of the fragment would compactly fold and occupy the extension. Prior studies have demonstrated that in *Avena* LOV2 with a Jα-helix, the helix is in close contact with the α/β-scaffold [17,34]. The length of the linker region in the recombinant fragment and buffer condition may alter the association with LOV2.

The molecular model of LOV1 indicated a volume twice that of the LOV2 model and suggested an anti-parallel association of LOV1 (Figure 3d). This anti-parallel association was confirmed by the crystal structure of phot2 LOV1 [18] (Figure 3d). The dimeric association could be attributed to the interactions of the Thr215 and Thr216 sidechains and associated hydration water molecules bound to the interface (Figure 3d). The structural evidence from SAXS and crystal structures indicated that phot2 likely exists as a dimer in solution with dimerization sites at LOV1.

### 2.3. Asp720Asn-Mutated LOV2-STK

In contrast to the LOV fragments described above, the SAXS profiles of Asp720Asn-mutated LOV2-STK (Figure 1a), which lacks phosphorylation activity, differed between D_450_ and S_390_ in the measured concentration range. The SAXS intensities of S_390_ decreased by approximately 6% compared to those in D_450_ at 0.004 < S < 0.012 Å^−1^ (Figure 4a). The findings suggested BL-induced conformational changes, such as rearrangement of domains separated by approximately 100 Å, according to the reciprocity between the scattering vector and distance of the electron pairs. The profiles after the dark reversion resembled those of D_450_ (Figure 4a). This finding indicated the photo-reversible nature of the BL-induced conformational changes in LOV2-STK (Figure 4a).

The Guinier plots of LOV2-STK under the three light/dark conditions produced approximately straight lines (Figure 4b), and the *C*/(*I*(*S* = 0,*C*) and *R*g(*C*)^2^ values displayed linear concentration dependencies (Figure 4c). Therefore, the LOV2-STK solutions were monodispersive in the measured concentration range. In addition, LOV2-STK existed as a monomer, as judged from the apparent (Table 1). The 2.4 Å increase of *R*g(*C* = 0) in S_390_ by from that in D_450_ (Table 1) implied an expansion and/or swelling in the molecular dimensions of LOV2-STK under BL irradiation. In contrast, the comparable *R*g(*C* = 0) values between the D_450_ and after dark reversion confirmed the photo-reversible conformational changes as well as the scattering profiles.

The D_450_ model is composed of a main body shaped like a rectangular plate and an ellipsoid-shaped, small lobe attached to the edge of the main body (Figure 4d). The restored molecular model of the D_450_ state almost overlapped with that in the dark reversion. The shapes and sizes of the small lobe and the main body were consistent with those of LOV2-Jα and a homology model of STK constructed from protein kinase A (PKA), respectively (Figure 4d). In addition, the main body resembled the overall shapes of Tyr and Ser/Thr kinases, such as Fes tyrosine kinase (Fes) [60] and C-terminal Src kinase (Csk) in their inactive and active forms [61], Anti-Ca/CaM kinase II-a-subunit (c-Abl) in the activated form [62], and Rio2 serine kinase (Rio2) [63]. By referring to the arrangements of kinase and regulatory domains in these kinase structures, the homology model of STK and LOV2 could be fitted to the main body and small lobe, respectively. In this arrangement, LOV2 occupied the position of regulatory domains in protein kinases and adopted an edge-to-edge association with the N-terminal domain of STK. The linker region may occupy the available space between the LOV2-Jα and STK models or may have large conformational fluctuations that do not contribute appreciably to the scattering intensity.

In the molecular model of S_390_, the small lobe appeared to have a more elongated shape than that of D_450_ (Figure 4d). Since the main body of D_450_ overlapped with that of S_390_, LOV2 moved by approximately 10 Å from the position in D_450_ in a BL-dependent manner with a slight inclination of the molecular axis. This positional shift of LOV2 could be a major cause of the changes in the SAXS profile, including increased *R*g(0) (Table 1). The domain organization and BL-induced rearrangement of domains are discussed below.

### 2.4. Asp720Asn-Mutated Full-Length Phot2

Conversion to S_390_ of LOV2 is essential for phot2 and the dark reversion speed of S_390_ to D_450_ is a key factor that enhances STK activity [64]. A BL fluence of more than 450 μmol m^−2^ s^−1^ converted almost all Asp720Asn mutated full-length phot2, and more than 90% of Cys170Ala/Asp720Asn and Cys426Ala/Asp720Asn mutants (which lost the adduct formation capability of LOV1 and LOV2, respectively) to their S_390_ state. In contrast, the Cys170Ala/Cys426Ala/Asp720Asn triple-mutant used as a reference displayed little change in its absorption spectrum under BL irradiation (Figure 5a).

In the dark reversion of Asp720Asn-mutated phot2, S_390_ of LOV2 and LOV1 decayed with a half-lifetime (t_1/2_) of 54 s and slowly with an t_1/2_ of 194 s, respectively. Interestingly, S_390_ of LOV2 in Cys170Ala/Asp720Asn-mutated phot2 decayed in a single-exponential mode with an t_1/2_ value of 168 s, which was approximately three times longer than that of LOV2 in Asp720Asn-mutated phot2. The findings indicated the structural influences of LOV1 on the dark reversion of LOV2. In addition, S_390_ of LOV1 in Cys426Ala/Asp720Asn-mutated phot2 displayed faster dark reversion, a single-exponential with an τ_1/2_ of 68 s, than that in Asp720Asn-mutated phot2. These findings indicated mutual structural influences of LOV1 and LOV2 on their photocycles.

SAXS profiles of Asp720Asn-mutated phot2 are shown in Figure 5b. The scattering intensities of S_390_ increased by more than 8% from that of D_450_ in S < 0.007 Å^−1^ (Figure 5c), while the profile after the dark reversion resembled that in the dark. Guinier analysis (Figure 5d) and the observed concentration dependence of *C*/*I*(*S* = 0,*C*) and *R*g(*C*)^2^ (Figure 5e) indicated the monodispersive nature of phot2 solutions. The *M*_w_ estimated from *C*/*I*(*S* = 0,*C* =0) confirmed the dimeric association of phot2. The value of D_450_ was comparable with that after dark reversion, but was slightly larger than that of S_390_. In contrast, the maximum dimensions were almost the same between D_450_ and S_390_ (Table 1). The changes in the scattering profile and *R*g value of S_390_ could be attributed to photo-reversible conformational changes of phot2, which produced a smaller molecular size of S_390_ compared to the size of D_450_. The surface properties of phot2 remain unchanged under BL irradiation, as indicated by the small changes in parameters *A*_2_ and *B*_if_ (Figure 5e).

For phot2, ab initio calculations yielded a variety of molecular models with respect to shape. The most probable models (Figure 5f) were selected using our protocol with multivariate analysis [59] (Figure 2d). The molecular model of D_450_ and that after the dark reversion displayed an approximately rod shape when viewed from the direction perpendicular to the two-fold symmetry axis, as shown in the lower row of Figure 5f. Cys170Ala/Cys426Ala/Asp720Asn-mutated phot2, which displayed little BL-induced conformational changes as demonstrated by SAXS measurement (Figure 5b), appeared similar to the SAXS model of D_450_ in transmission electron microscopy (TEM) imaging [43]. The model of S_390_ bent near the center of the subunit to swing both edges of the rod by approximately 30 Å. The bent shape in S_390_ could explain the slightly smaller than that in D_450_. The LOV1 dimer occupied the center of the shape, and the LOV2-STK model in D_450_ (Figure 4d) fit both edges of D_450_ and S_390_ (Figure 5f). The details of the arrangements of the functional domains are discussed below.

SAXS measurements for mutants lacking the adduct formation capability of LOV1 (Cys170Ala/Asp720Asn-mutant) or LOV2 (Cys426Ala/Asp720Asn-mutant) provided clues to understand which BL-activated LOV domain was responsible for the BL-induced conformational changes in Asp720Asn-mutated phot2. Both Cys170Ala/Asp720Asn and Cys426Ala/Asp720Asn mutants displayed significant SAXS differences in the dark and under BL irradiation to a lesser extent than in the Asp720Asn mutant (Figure 5c). This finding implied that each of the BL-excited LOV1 and LOV2 could induce conformational changes that were detectable by SAXS, but that both BL-excited LOV domains were required to induce structural changes that were necessary for the SAXS differences in the Asp720Asn mutant. Therefore, BL-excited LOV1 would partly and structurally contribute to enhanced STK activity.

## 3. Discussion

The SAXS data on recombinant fragments and full-length phot2 from A. thaliana revealed undetectable conformational changes in BL-excited LOV1, LOV2, and LOV2-linker fragments. However, SAXS changes explained by rearrangements of functional domains were observed in LOV2-STK and full-length phot2. Here, we discuss the implications of the molecular models from the SAXS data and results from biochemical assays and spectroscopic measurements to illustrate the molecular organization and intramolecular mechanism of the induction of cellular signaling cascades in the BL-response by phot2.

### 3.1. Structures of LOV1, LOV2, and LOV2-Linker

Recombinant LOV1 alone exists as a homodimer in solution [41] and as a crystal [18] (Figure 3d). In both SAXS and crystallization solutions, LOV1 concentrations are much higher than those of phot2 in vivo. However, because phot2 forms dimer in diluted conditions, as observed by TEM [43], the LOV1 domain must act as a dimerization site of phot2. LOV1 of *Arabidopsis* phot1 forms dimers by a disulfide bridge [18]. In *Arabidopsis*, the phot2 LOV1-related LOV domain from Flavin-binding, Kelch repeat, F-box protein (FKF1) is a photoreceptor that regulates flowering. SAXS data have demonstrated FKF1 dimers in solution [65]. In addition, FKF1-related ZEITLUPE [66] and LOV Kelch protein2 [67] are also expected to form dimers at their N-terminal LOV domains based on their sequence similarities. Therefore, in photoreceptor proteins of *Arabidopsis*, LOV1 may have dual functional roles as a photo-receiving unit and a dimerization site. Since homo- or hetero-dimerization of PAS superfamily proteins is crucial for cellular signal transduction [68,69,70], LOV1 of *Arabidopsis* photoreceptor proteins, a subset of the PAS superfamily, may have acquired the capability of dimerization during the evolution of *Chlamydomonas* phot, which exists as a monomer [33].

Although only LOV2 has been thought to have a predominant role in BL-dependent enhancement of phosphorylation activity of STK, the BL-induced SAXS change of Cys426Ala/Asp720Asn-mutated phot2 (Figure 5c) suggests a certain structural influence of BL-activated LOV1 on the rearrangements of domains in phot2 in S_390_, although LOV1 itself may have little direct contact with STK in the tandem configuration of LOV1, LOV2, and STK (Figure 5f). The structural contribution of BL-induced LOV1, which exhibits very small changes in the SAXS profile, to the rearrangements of functional domains in phot2, are discussed below.

Recombinant LOV2 and LOV2-linker fragments are monomers in solution, while those of phot1 form dimers [20]. The differences arise from the surface properties of LOV2 between the two isoforms. Residues exposed to solvent in phot1 are taken place by residues with opposite electrostatic properties in phot2. In addition, 15 residues in the linker of phot2 contains only two negatively charged residues, whereas five residues are present in phot1.

The LOV2 and LOV2-linker of phot2 display very small SAXS changes between D_450_ and S_390_. In fact, S_390_ of LOV2-Jα displays conformational changes that were too small to be detectable by X-ray crystallography and nuclear magnetic resonance chemical shifts. As described below, the linker region, including the Jα-helix acts as an amplifier for the small conformational changes of LOV2 and therefore subsequently triggers the activation of STK.

### 3.2. Molecular Shape and BL-Induced Rearrangement of Domains in LOV2-STK

Asp720Asn-mutated LOV2-STK is monomer in solution (Table 1). The data informed the proposed molecular models of D_450_ and S_390_. In the D_450_ model shown in Figure 4d, LOV2 is putatively placed on the small lobe. However, its orientation relative to STK is unclear. To explore the contacts between LOV2 and STK, the interaction modes of regulatory and kinase domains in c-Abl [62] and Rio2 [63] kinases provides a clue (Figure 6a). In each kinase, the regulatory domain (Src homology 2 domain [SH2] in c-Abl or the winged-helix domain [WHD]) touch the curved anti-parallel β-sheet in the *N*-domain. Particularly, in Rio2 kinase, one helix of WHD associates with the shallow groove of the β-sheet. In the putative model (Figure 6b), LOV2-Jα is placed on the β-sheet of the STK *N*-domain in the same manner as the helix of Rio2 WHD (Figure 6b). In this association, αA/β1 in the *N*-terminal region of LOV2, which is crucial for the activation of STK [71], is located near the *N*-domain of STK.

The BL-induced SAXS changes (Figure 4a–c and Table 1) indicates the photo-reversible movement of LOV2 relative to STK (Figure 4d). In contrast, structural changes of LOV2 alone in S_390_ are undetectably small in SAXS (Figure 3a,b). In addition, the observations that the main body of the SAXS models are nearly identical between D_450_ and S_390_. Therefore, the positional shift of LOV2 could be attributed to the conformational alternation of the linker region, including the Jα-helix. BL-triggered conformational changes and/or unfolding of the Jα helix have been proposed based on studies of *Avena* phot1 LOV2 and *Arabidopsis* phot2 LOV2 [34,35,36,50,74]. As the structural flexibility of the linker region is expected from the SAXS models of the LOV2-linker (Figure 3c) and LOV2-STK models (Figure 4d), the linker region would be able to structurally adapt to external perturbation. In a PAS kinase, the *N*-terminal PAS domain functions as a ligand-regulated molecular switch that induces kinase activity [60]. In an analogy, the linker region may also function as a regulator or a molecular switch amplifying the light-induced structural changes in LOV2 to the kinase domain.

Similar BL-induced conformational changes have been observed in SAXS studies for recombinant LOV2-STK fragments of *Arabidopsis* phot1 [32] and *Chlamydomonas* phot [33]. Therefore, BL-induced reorganization of LOV2 and STK is common among *Arabidopsis* phot1 and phot2, and *Chlamydomonas* phot. In phot1, Lys475, Lys636, and adjoining residues in LOV2-STK fragments are essential to regulate the propagation of structural changes from BL-activated LOV2 to STK [32].

The SAXS profile and the molecular shape of LOV2-STK in D_450_ (Figure 4a,d) do not support a proposed scheme for the enhancement of phosphorylation activity of STK by BL-activated LOV2. In this proposal, LOV2 directly associates with the active-site cleft of STK in D_450_, similar to the regulatory subunit RIα associated with PKA [74,75], and BL-activated LOV2 dissociates from the cleft for substrate access. However, the simulated SAXS profiles from the proposed LOV2-STK model are markedly different from the experimental profiles of LOV2-STK [76].

### 3.3. Organization and BL-Induced Rearrangement of Domains in Phot2

In the SAXS model of phot2 in D_450_, the LOV1 dimer acts as a subunit association site and occupies the center of the dimer, and LOV2 and STK are tandemly arranged toward the edge of the model (Figure 6c) as those in the LOV2-STK model of D_450_ (Figure 4d). In this arrangement, the active-site cleft of STK is distant from most of the phosphorylation sites identified in the *N*-terminal half (Figure 1a) [43]. Therefore, most of the sites would be modified by another phot2. In contrast, only Ser16 and Ser22 at the edge of the 100-residue long *N*-terminal tail may touch the cleft of STK for auto-phosphorylation.

The molecular shape of S_390_ is similar to that of D_450_ when viewed along the direction perpendicular to the two-fold axis but bends in the middle of the LOV1-LOV2-STK rods of subunits (Figure 5f). Since the LOV1 dimer occupying the central part exhibits little conformational change [18], the bends are probably attributed to the reorganization of LOV2 and STK, as expected from the BL-induced structural changes in the LOV2-STK fragments of *Arabidopsis* phot2 (Figure 4d) [22], phot1 [32], and *Chlamydomonas* phot [33]. The BL-activated LOV2 domain triggers conformational changes of the linker that can separate the two domains, as speculated in the previous section regarding the activation mechanism of LOV2-STK (Figure 4d) [22]. In contrast to LOV2-STK, the LOV1 domain in full-length phot2 would mechanically suppress the dissociation of LOV2 from STK. Then, LOV2 and STK will rearrange, rather than separate, at their interface to accommodate structural changes in the Jα and αA/β1 regions. Both regions appear to be crucial for the intramolecular transduction of structural changes [34,37]. In addition, in the LOV2-STK of phot1, the *N*- and *C*-terminal regions of LOV2 are critical for transmitting conformational changes of LOV2 to STK [32,71].

In the model of D_450_ (Figure 6b), LOV1 and LOV2 are placed so that their FMN chromophores are separated by approximately 18 Å (Figure 6b). In addition, due to the loop contacts near β1 of the *N*-terminal region of LOV1 with LOV2, the conformational changes in LOV1 may influence LOV2. Therefore, in the proposed model, LOV1 could structurally influence the photosensitivity in the BL-dependent kinase activation, probably by modifying the lifetime of S_390_ in LOV2 [64]. Therefore, in LOV1 and LOV2, even small conformational changes exert mutual influences on their photocycles. Changes in the SAXS profile of Cys426Ala/Asp720Asn mutant under BL irradiation are comparable with the changes of the Cys170Ala/Asp720Asn mutant and are probably caused by less pronounced bends than in the Asp720Asn mutant (Figure 5c). This finding implies that small conformational changes in LOV1 contribute to the conformational changes of phot2, and that LOV1 and LOV2 cooperatively contribute to structural changes.

In addition to SAXS data, biochemical assay data have also provided evidence of the involvement of LOV1 in the activation of STK. For instance, Cys426Ala-mutated phot2 cannot form S_390_ but retains kinase activity in vitro [30] and in vivo [13,77]. In *Chlamydomonas* phot, in which LOV1, LOV2, and STK are in a tandem arrangement like the subunit of the phot2 dimer, LOV1 is essential to enhance the sensitivity of the fluence-dependent response [33].

Here we have revealed the dimer formation of phot2 and the most probable organization of functional domains from the SAXS data [43]. We also proposed the domain organization of phot1 dimer from *Arabidopsis* [32]. Therefore, both BL-activated phot1 and phot2 can simultaneously phosphorylate two protein molecules in the signal transduction pathway. This would be advantageous to increase the signaling cascades concerning with photosynthesis optimization in plants.

### 3.4. Future Structural Studies of Phot2

In the present study, to ensure the homogeneity of recombinant proteins in physicochemical measurements, we introduced the Asp720Asn mutation to abrogate the affinity to Mg-ATP. However, because phosphorylation may still influence BL-induced structural changes in wild-type phot2, purification techniques to separate wild-type phot2 at different phosphorylation levels need to be developed to understand the influences of phosphorylation on conformational changes of phot2.

Recently, we began cryogenic TEM observations of phot2 [43]. Cryogenic TEM can classify images of heterogeneous structures of proteins caused by chemical modification, intrinsic molecular motions, and/or BL-induced conformational changes [78]. However, the stabilization buffer of phot2 used in SAXS experiments is unsuitable for cryogenic TEM observation of frozen-hydrated phot2 because of the high concentrations of NaCl and glycerol. In addition, diffusion of phot2 during flash-cooling probably causes denatures phot2 at the buffer-air interface, as observed in crystallization trials. Therefore, buffer solutions and preparation procedures that avoid interfacial denaturation must be searched for structure analyses using cryogenic TEM.

## 4. Materials and Methods

### 4.1. Specimen Preparation

The expression and purification of recombinant LOV1, LOV2, LOV2-linker, Asp720Asn-mutated LOV2-STK fragments, Asp720Asn-, Cys170Ala/Asp720Asn-, Cys426Ala/Asp720Asn-, and Cys170Ala/Cys426Ala/Asp720Asn-mutated full-length *Arabidopsis* phot2 have been reported [22,41,43]. In both the LOV2-STK fragment and full-length, Asp720Asn mutations that abrogate the affinity of STK to Mg-ATP, are introduced to avoid heterogeneous auto-phosphorylation in specimen purification [22,43]. Cys170Ala and Cys426Ala mutations cause the loss of the adduct formation capability of LOV1 and LOV2, respectively.

Briefly, each DNA fragment encoding a fragment or full-length was inserted into an overexpression vector, and the vector was transformed into *Escherichia coli* (Sigma-Aldrich, Tokyo, Japan). Expression of the inserted DNA was induced by adding isopropyl β-d-1-thiogalactopyranoside (Wako, Osaka, Japan) to the culture in the dark. Cell harvest and purification were performed under dim red light. After centrifuging the crude extract, fragments and full-length were purified by combining affinity chromatography, SEC, and anion exchange chromatography and SEC. The purity of the specimens was examined using sodium-dodecyl-sulphate polyacrylamide gel electrophoresis after each column chromatography step. The purity of each purified specimen solution exceeded 98%. Prior to SAXS measurements, the apparent molecular weight of each purified protein was estimated by elution with SEC.

Absorption spectra and the time course of absorption changes in photo-conversion from D_450_ to S_390_, and in the dark reversion from S_390_ to D_450_ were measured at 293 K using a spectrophotometer. BL irradiation of specimens to accumulate S_390_ was carried out using a BL emitting diode with an emission maximum at 450 nm. The fluence rate necessary to convert almost all molecules in each specimen to S_390_ was determined.

### 4.2. SAXS Measurements

SAXS profiles were collected at the BL40B2 and BL45XU SAXS beamlines of SPring-8 at 293 K (Figure 2a). The X-ray wavelength was 1.0000 Å. A single specimen cell with quartz windows of 0.01 mm-thickness and a 3.0 mm path length was used. In the early stages of this research, R-axis IV (RIGAKU, Japan) was used as an area detector. Subsequently, PILATUS detectors (DECTRIS, Baden, Switzerland) were used. The specimen-to-detector distance was 2.0 to 2.5 m.

SAXS profiles of each specimen were collected at least three specimen concentrations to estimate structural parameters under infinitely diluted conditions. For each specimen solution, we collected SAXS sequentially in the dark (D_450_), under BL irradiation (S_390_), and after more than 15 min in the dark (dark reversion from S_390_ to D_450_). A BL emitting diode was placed near the specimen solution so that the fluence rate was sufficient to convert almost all molecules in the specimen solution to S_390_ and to accumulate S_390_ during X-ray exposure (Figure 2a). The absorption spectra and SDS-PAGE patterns after X-ray exposure as well as the stabilities of SAXS profiles confirmed that there was negligible radiation damage.

### 4.3. SAXS Analysis

A one-dimensional SAXS profile was obtained by circular averaging of the two-dimensional recorded SAXS pattern after subtracting the background scattering obtained from a buffer solution (Figure 2b). The SAXS profile in a very small-angle region was analyzed by Guinier’s approximation (Figure 2c) [79]. For this, the SAXS intensity at scattering vector length *S* from a solution of protein concentration *C*,/*I*(*S*,*C*), was approximated as seen in Equation (1):(1)$$I\left(S,C\right)=I\left(S=0,C\right)\times \exp\left[-4\pi^{2}\times \frac{Rg^{2}\left(C\right)S^{2}}{3}\right]$$
where *S* = 2 × sin*θ*/*λ*, *I*(*S* = 0,*C*) and *R*g(*C*) are the forward scattering intensity and radius of gyration at protein concentration *C*, respectively. The scattering vector length *S* is defined by the scattering angle, 2*θ*, and the wavelength of the X-ray, *λ*.

Under the dilution condition, *I*(*S* = 0,*C*) and *R*g(*C*) depend on the concentration (as shown in Equations (2) and (3)):(2)$$KC/I\left(S=0,C\right)=\frac{1}{M_{W}}+2\times A_{2}\times C$$
(3)$$Rg^{2}\left(C\right)=Rg\left(C=0\right)-B_{if}\times C$$
where *M*_w_ is the apparent molecular weight of the protein, *K* is an experimental constant, *A*_2_ is the second virial coefficient, and *B*_if_ reflects the mode of intermolecular interactions [79]. The signs of *A*_2_ and *B*_if_ are usually the same. When the SAXS profiles in a concentration range and the obtained parameters satisfy Equations (1), (2) and (3), respectively, the specimens in solution are monodispersive. The maximum dimension of a protein is estimated from the distance-distribution function calculated using the GNOM program [80].

### 4.4. Molecular Shape

Molecular models were restored as an assembly of dummy residues (DR) with a diameter of 3.8 Å by the GASBOR program [55], which minimizes the discrepancy between the experimental (*I*_exp_(*S*)) and calculated (*I*_model_(*S*)) scattering profiles under a restraint regarding a compact packing of DRs. The discrepancy between *I*_exp_(*S*) and *I*_model_(*S*) is monitored by the χ^2^-value defined as in Equation (4):(4)$$\chi^{2}=\sum {\left[\frac{\left\{I_{exp}\left(S_{j}\right)-K\times I_{model}\left(S_{j}\right)\right\}}{\sigma \left(S_{j}\right)}\right]}^{2}/\left(N-1\right)$$
where *K* is a scale factor, and *S*_j_ is the scattering vector length of the j-th data point among experimental data. *σ*(*S*_j_) is the error in the measured intensity profile. χ^2^-value depends on the number of DRs. Thus, the optimum number of DRs yielding the smallest χ^2^ was estimated from multiple trial calculations with different numbers of DRs, as reported previously [41,43]. Since the optimum number of DRs is affected by the electron density contrast between the protein molecule and buffer solution [81], the optimum number of DRs is frequently smaller than the number of amino acid residues of a protein [56]. It should be noted that mirror images of restored models can also provide SAXS profiles to explain the experimental one.

When models restored from a SAXS profile with a high ambiguity score [82] are in various shapes and sizes, we apply our multivariate protocol to a set of restored models (Figure 2d) [59]. After superimposing models onto a reference with respect to their moments of inertia, the distribution of DRs in each model digitized into an array of 4 × 4 × 4 Å^3^ voxels. Each model is expressed as a point in a multi-dimensional space spanned by a set of voxels. Principal component analysis was performed to reduce the distribution of models in this multi-dimensional space onto the plane spanned by the first and second principal components. Classes of probable models were selected after K-means clustering and molecular models were obtained by averaging models in the selected classes.

## 5. Conclusions

We conducted SAXS studies of *Arabidopsis* phot2, which regulates responses for efficient photosynthesis. Phot2 is composed of LOV1 and LOV2 domains for BL absorption and STK for intermolecular signal transmission by phosphorylation. We investigated the structural mechanism of intramolecular signal transduction, which converts the BL stimuli to biochemical signals in phot2. In the dark, phot2 formed an S-shaped dimer by LOV1 with the tandem arrangement of LOV1, LOV2, and STK in each subunit. LOV2 was located near the N-terminal of STK, similar to other kinase inhibitor proteins.

Under BL irradiation, LOV1 and LOV2 were converted from D_450_ to S_390_. Although LOV1 did not directly regulate the activity of STK, LOV1 controlled the photosensitivity of the kinase activity of STK by modifying the lifetime of LOV2 at S_390_. Photo-conversion and subsequent small structural changes of LOV2 produced conformational changes in the linker region between LOV2 and STK, such as unfolding of the Jα helix. Finally, domain rearrangements of LOV2 and STK occurred and were apparent as bending motions. The structural analyses demonstrate that these BL-induced sequential structural changes are the mechanism of BL-induced signal transduction of phot2. In addition, phot2 dimer, which can simultaneously phosphorylate two protein molecules in the signal transduction system under BL, would be advantageous to increase the signaling cascades concerning with photosynthesis optimization in plants. It is expected that future analysis will provide more details of this mechanism.

## Figures and Tables

**Figure 1 ijms-21-06638-f001:**
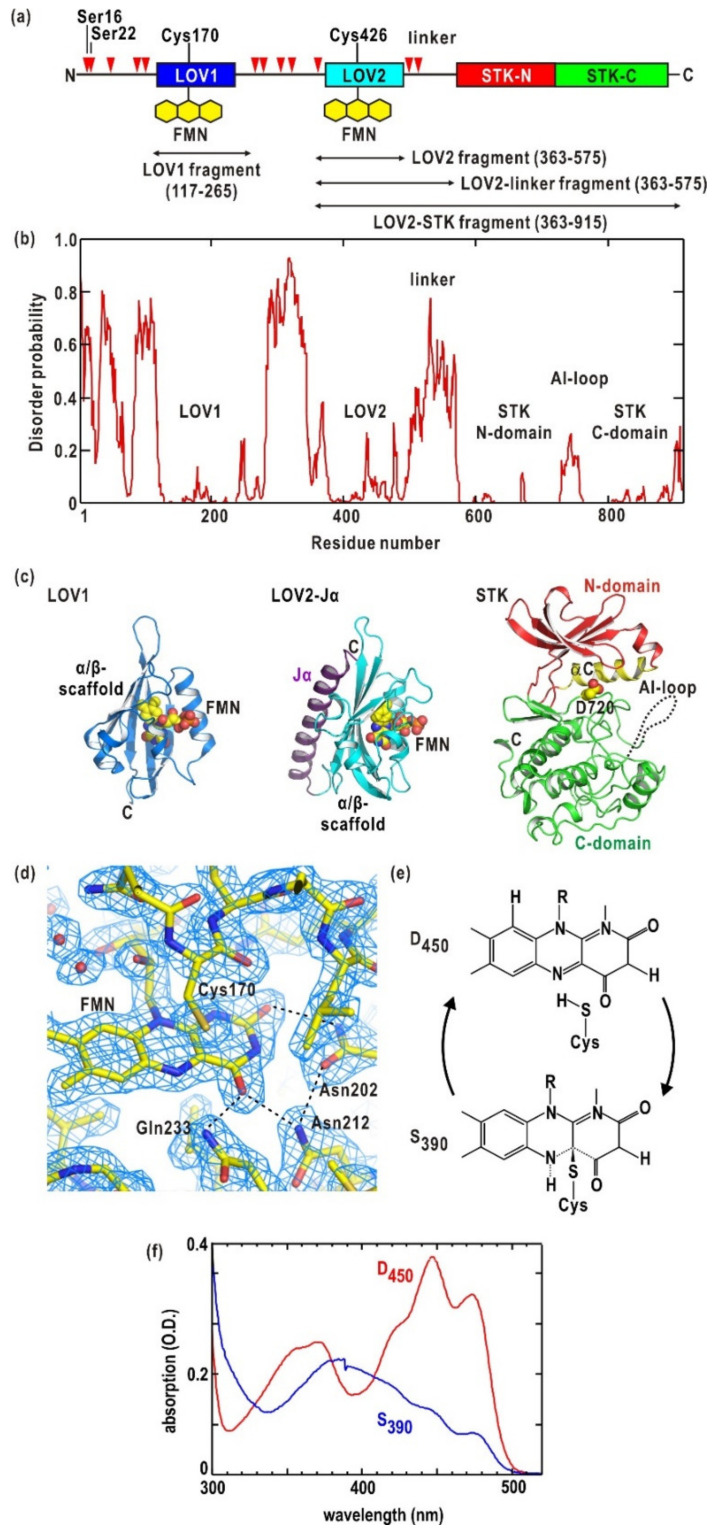
(**a**) The domain organization in *Arabidopsis* phot2. Fragments used in SAXS experiments are indicated. The red triangles denote phosphorylation sites. Some residues are labeled. (**b**) Disorder probability calculated by the DISOPRED2 program [41] plotted against residue number. Regions fold into functional domains and loops are labeled. (**c**) Protomer of LOV1 dimer (Protein Data Bank (PDB) accession code: 2Z6D) [18], homology models of LOV2 with Jα-helix and STK [42] illustrated as ribbon models. The *N*-, *C*-terminal domains, and α*C*-helix of STK, whose locations are important for phosphorylation activity, are colored differently. The predicted AI-loop is indicated by a dotted line. Residue Asp720 shown as a space-filling model is the point mutation site in Asp720Asn-mutated LOV2-STK and full-length. Hereafter, structure models were drawn by PyMol [43]. (**d**) Electron density map around FMN in a crystal structure of LOV1 dimer determined at a resolution of 2.0 Å [18]. FMN and some important residues in photo-reaction are labeled. The dashed lines indicate hydrogen bonds in these residues. (**e**) Schematic illustration of the photo-reaction cycle of the LOV domain. (**f**) Absorption spectra of D_450_ (red line) and S_390_ (blue) states of Asp720Asn-mutated phot2.

**Figure 2 ijms-21-06638-f002:**
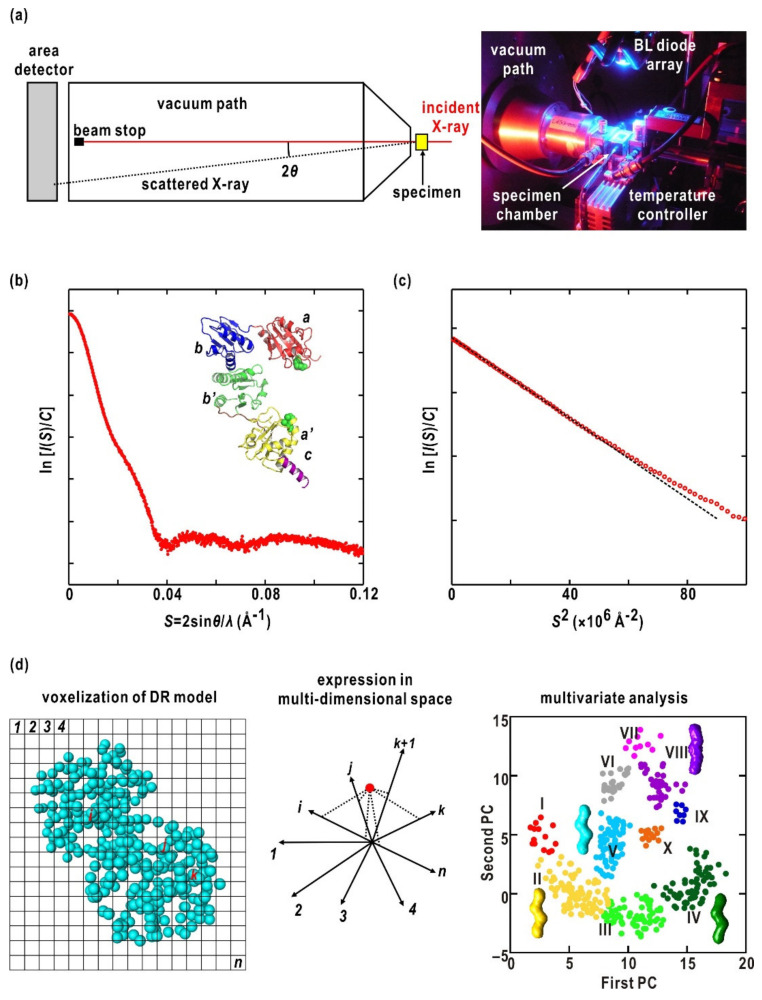
(**a**) A schematic illustration of the SAXS experiment (left) and a photograph of the specimen chamber (right). In the right photograph, the phot2 specimen was photo-excited by BL irradiation of a photo-diode. (**b**) SAXS profile of protein disulfide isomerase (PDI) from a thermophilic fungus [44]. A model of PDI is shown in the inset. The enhancements at approximately *S* = 0.03 Å^−1^ are caused by the characteristic arrangement of the four domains in PDI. (**c**) A Guinier plot for the small-angle region in the SAXS profile in panel (**b**). The dashed regression line indicates *R*g of 33.4 ± 0.4 Å. (**d**) Schematic illustration of the multivariate analysis of models restored by ab initio calculation by the GASBOR program [45]. The details are described in the Materials and Methods section.

**Figure 3 ijms-21-06638-f003:**
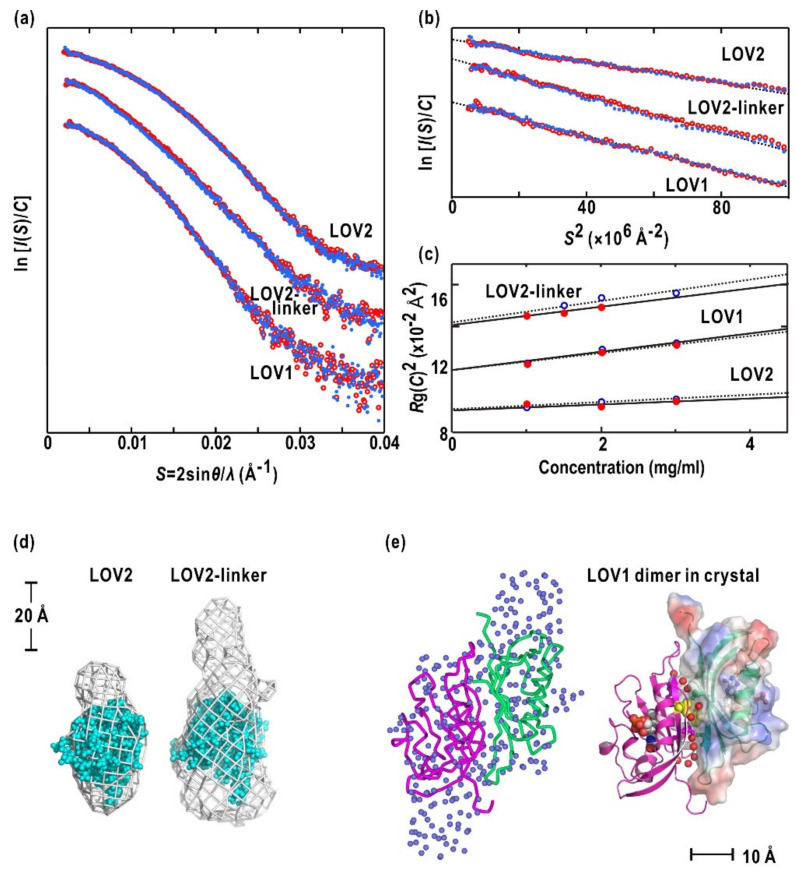
(**a**) SAXS profiles of LOV2 (top), LOV2-linker (middle), and LOV1 (bottom). The red circles and blue dots are profiles of D_450_ and S_390_, respectively. The concentration of each specimen is 3 mg mL^−1^. (**b**) Guinier plots for LOV2, LOV2-linker and LOV1. (**c**) Concentration-dependent variation of *R*g(C)^2^ values for LOV2-linker, LOV1, and LOV2 in the D_450_ (red circles) and S_390_ (blue) states. The linear concentration-dependences are indicated by black lines for D_450_ and dashed lines for S_390_. (**d**) Molecular shapes of LOV2 and LOV2-linker averaged from ten restored models. The LOV2 model shown as space-filling model is fit to the main body of each molecular shape. (**e**) Superimposition of a restored SAXS model (an assembly of blue spheres) onto the crystal structure of LOV1 dimer [18]. The right panel shows a crystal structure of LOV1 dimer determined at a resolution of 2.0 Å (PDB accession code: 2Z6D) [18]. The models of the polypeptides are shown as ribbon models. FMN molecules are shown as a space-filling model in the left protomer, and as a stick model in the right protomer. Yellow-colored space-filling models and red spheres are Thr217 residues and hydration water molecules, both of which predominantly contribute to the dimer formation. The solvent-accessible surface of the right protomer is illustrated. Red- and blue-colored surfaces indicate the positions of negatively- and positively-charged sidechains of amino acid residues, respectively. Panels (**a**) and (**b**) are reused with modification from reference [41] with permission from American Chemical Society. The left panel in (**d**) is reused with modification from reference [18] with permission from Elsevier.

**Figure 4 ijms-21-06638-f004:**
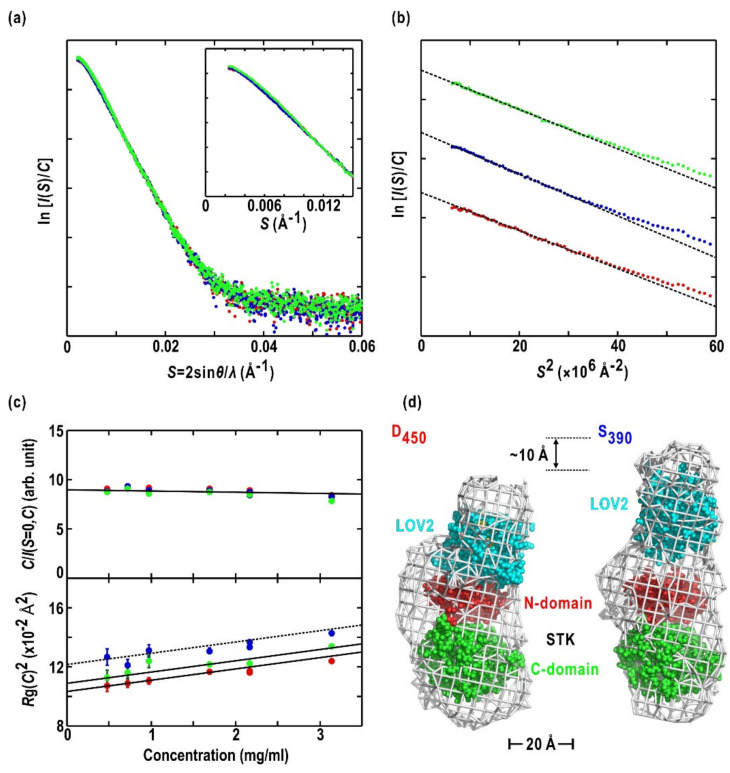
(**a**) SAXS profiles of LOV2-STK in D_450_ (red dots), S_390_ (blue), and dark reversion for 15 min (green). The inset shows a magnified view of the profiles to demonstrate the intensity decrease of S_390_ in 0.002 < S < 0.010 Å^−1^. The concentration of the LOV2-STK solution is 2.2 mg/mL. (**b**) Guinier plots for the profiles shown in panel (**a**). Plotted data are appropriately shifted for clarity. (**c**) Concentration-dependent variations of *I*(*S*,*C*) (upper panel) and *R*g(*C*)^2^ (lower panel). Each concentration dependence is approximated by linear regression lines. (**d**) Comparison of SAXS models of D_450_ (left panel) and S_390_ (right). LOV2-Jα and STK shown as space-filling models are fit to the averaged molecular shapes. Panels are reused with modification from reference [22] with permission from American Chemical Society.

**Figure 5 ijms-21-06638-f005:**
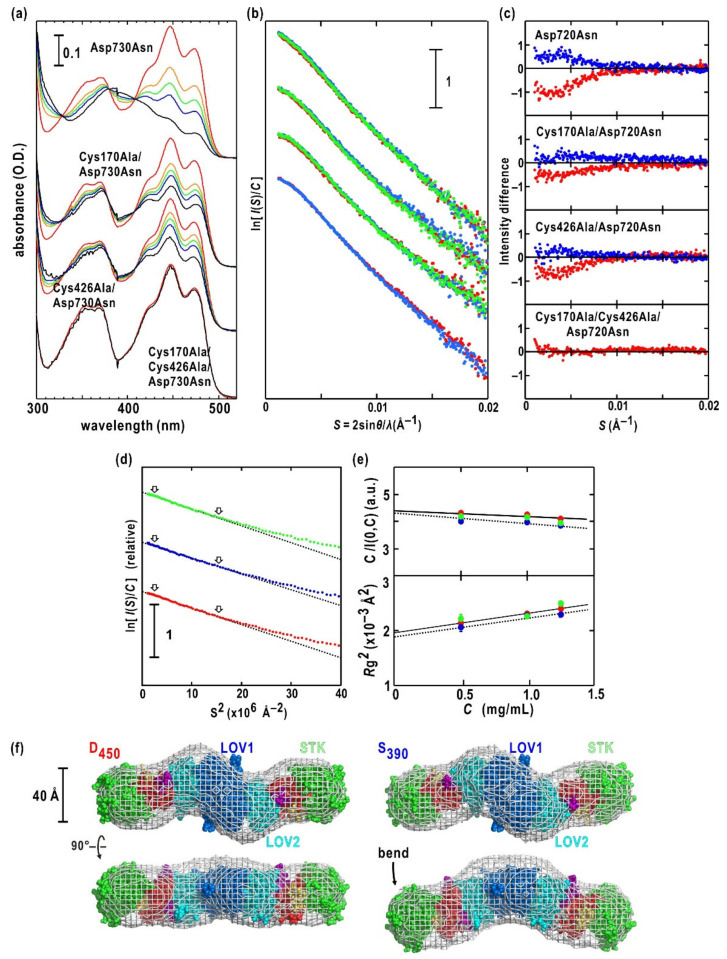
(**a**) Absorption spectra of Asp720Asn-, Cys170Ala/Asp720Asn-, Cys426Ala/ Asp720Asn-, and Cys170Ala/Cys426Ala/Asp720Asn-mutated phot2. The red and black lines show the D_450_ and S_390_ states, respectively. The orange, green, blue, and black (S_390_) spectra are recorded under BL irradiation (λ_max_ at 450 nm) at a fluence of 10 (orange), 25 (green), 50 (blue), and 200 (black) μmol m^−2^ s^−1^. (**b**) SAXS profiles of (from upper to lower) Asp720Asn-, Cys170Ala/Asp720Asn-, Cys426Ala/Asp720Asn-, and Cys170Ala/Cys426Ala/Asp720Asn-mutated phot2. The SAXS profiles of D_450_, S_390_ (blue), and nearly D_450_ after the dark reversion are indicated by red, blue and green dots, respectively. The concentration of each specimen is 0.8 mg mL^−1^. (**c**) Differences in intensities between D_450_ and S_390_ (blue dots) and between S_390_ and nearly D_450_ after the dark reversion (red) in Asp720Asn-, Cys170Ala/Asp720Asn-, Cys426Ala/Asp720Asn-, and Cys170Ala/Cys426Ala/Asp720Asn-mutated phot2. (**d**) Guinier plots of D_450_ (red dots), _S390_ (blue), and nearly D_450_ after the dark reversion (green) of Asp720Asn-mutated phot2. Regression lines determined by the Guinier approximation in the region are indicated by arrows. (**e**) Concentration dependencies of *C*/*I*(*S*,*C*) (upper panel) and *R*g(C)^2^ (lower) of D_450_ (red filled-circles), S_390_ (blue), and nearly D_450_ after the dark reversion (green) of Asp720Asn-mutated phot2. (**f**) Averaged molecular shapes of D_450_ (left panel) and S_390_ (right) as the most probable sets of molecular models selected using the multivariate protocol [59]. Space-filling models are the crystal structure of LOV1 dimer (PDB accession code: 2Z6D) [18] (blue colored), homology models of LOV2-Jα (cyan), and STK (red for the N-domain and green for the C-domain). The models are viewed from two different directions to demonstrate the bends of the molecular shape in S_390_. Panels are reused with modification from reference [43] with permission from American Society for Biochemistry and Molecular Biology.

**Figure 6 ijms-21-06638-f006:**
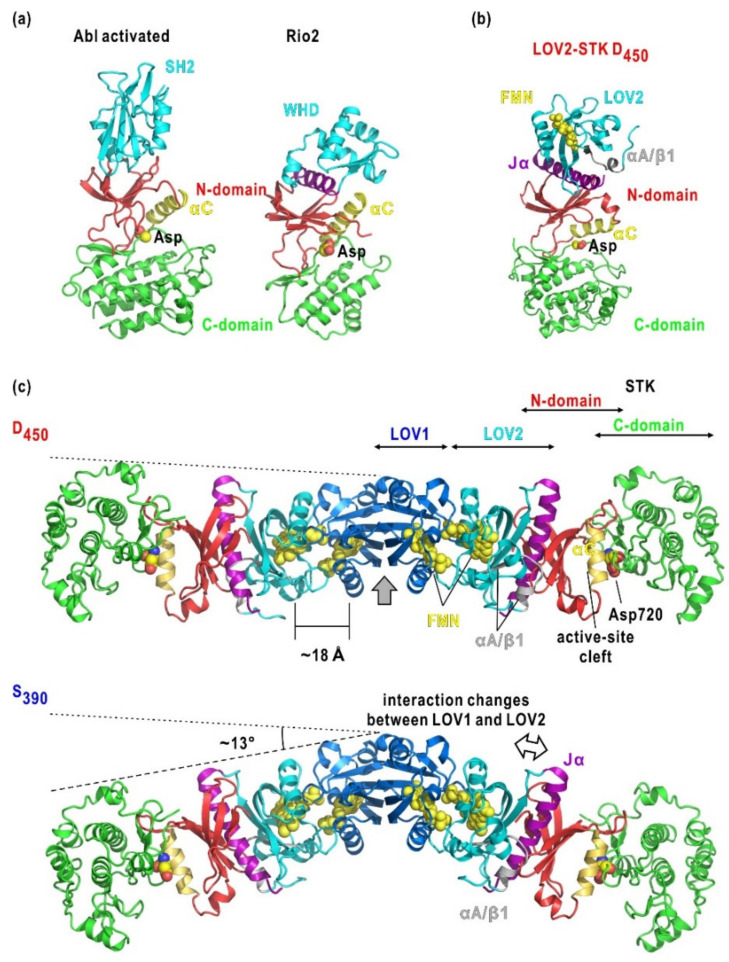
(**a**) Structures of c-Abl kinase in the activated state (PDB accession code: 1OPL) [72] and Ser kinase Rio2 (PDB accession code: 1ZAO) [73]. The regulatory domain (SH2 or WHD), *N*-terminal and *C*-terminal parts of the kinase domain are colored cyan, red, and green, respectively. Helix α*C* tuning the structure of the active-site cleft is colored yellow. In WHD, an α-helix contact with the β-sheet in the *N*-terminal part of the kinase domain is colored purple. The aspartate residues necessary for binding Mg-ATP is shown as a sphere model. (**b**) Model of LOV2-STK in D_450_ illustrated and viewed as in panel (**a**). (**c**) Domain arrangement in phot2 dimer in D_450_ and S_390_. Domains are shown in the coloring scheme in Figure 1. Some parts are labeled. The arrow indicates the two-fold rotation axis in the dimer. Dashed lines connecting the edges of LOV1 and STK are drawn to demonstrate the degree of the bend.

**Table 1 ijms-21-06638-t001:** Structural parameters of recombinant fragments and full-length of phot2.

Fragment (Mutation)	Residue Start/eEd	Oligomeric State	*R*g Dark/Light (Å)	*D*_max_ Dark/Light (Å)
LOV1 [18,41]	117/265	dimer	22.2 ± 0.3/22.6 ± 0.3	94 ± 3/94 ± 3
LOV2 [41]	363/500	monomer	20.2 ± 0.2/20.1 ± 0.3	72 ± 2/72 ± 2
LOV2-linker [41]	363/575	monomer	24.3 ± 0.3/25.0 ± 0.4	107 ± 3/110 ± 3
LOV2-STK [22] (Asp720Asn)	363/915	monomer	32.4 ± 0.4/34.8 ± 0.7	133 ± 2/140 ± 2
Full-length [43] (Asp720Asn)	1/915	dimer	44.3 ± 1.4/43.7 ± 1.4	188 ± 2/186 ± 2

The molecular weight of the specimen protein was estimated by referencing the *I* (*S* = 0, *C* = 0) value of hen egg-white lysozyme that has a known *M*_w_ or scattering from the buffer solution. When referring to the SAXS of lysozyme, the apparent *M*_w_ of a target protein was estimated using the partial specific volume of 0.74 cm^3^/g for soluble proteins.

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
