# Peer review of "Domain Organization in Plant Blue-Light Receptor Phototropin2 of Arabidopsis thaliana Studied by Small-Angle X-ray Scattering"

_ijms, 2020, doi:10.3390/ijms21186638_

Round 1

Reviewer 1 Report

The study of Nakasako et al. describes the reorganization of light-oxygen-voltage sensing domains in phototropin2 protein using SAXS.

The study is very comprehensive and was performed carefully and methodologically and is presented well in the manuscript.

Some minor revision is required for spelling correction and grammar, for example: line 20 is written 'LOV2-liker' instead of LOV2-linker.

line 95-96 'as do as' instead of ad do...

line 173 ends with the word 'and'

Overall it is a very good work and I recommend it for publication.

Author Response

First of all, the authors thank to their valuable comments on our previous manuscript. We corrected spelling and grammar throughout the revised manuscript. Although our previous manuscript was edited by a native speaker before submission, equations used were lost during the format change at the Editorial office. Then, many sentences had grammatical errors. In the revised manuscript, we inserted equations (1)-(3) as lines 524-550, and physical variables used throughout the main text. Spelling check were carefully checked throughout the revised manuscript. ‘LOV2-liker’, lines 95-96 and 173 were corrected as lines 20, 95, and 172, respectively. In addition, the references were rearranged according to the arrangement of Figures after the format changes by the Editorial office.

Reviewer 2 Report

Data were presented well, however at the end of discussion or at conclusion, please add a short outlook regarding the implications of these findings to the process of photosynthesis.

Author Response

First of all, the authors thank to their valuable comments on our previous manuscript. We added a short outlook on the implications of the present results as lines 464-469 in the Discussion section and lines 580-583 in the Conclusion section.